# Modification of Polymeric Carbon Nitride with Au–CeO_2_ Hybrids to Improve Photocatalytic Activity for Hydrogen Evolution

**DOI:** 10.3390/molecules27217489

**Published:** 2022-11-03

**Authors:** Linzhu Zhang, Lu Chen, Yuzhou Xia, Zhiyu Liang, Renkun Huang, Ruowen Liang, Guiyang Yan

**Affiliations:** 1Province University Key Laboratory of Green Energy and Environment Catalysis, Ningde Normal University, Ningde 352100, China; 2Fujian Provincial Key Laboratory of Featured Materials in Biochemical Industry, Ningde Normal University, Ningde 352100, China

**Keywords:** CeAu–CN heterostructure, interfacial interaction, interior electronic transmission channel, photocatalytic hydrogen production

## Abstract

The construction of a multi-component heterostructure for promoting the exciton splitting and charge separation of conjugated polymer semiconductors has attracted increasing attention in view of improving their photocatalytic activity. Here, we integrated Au nanoparticles (NPs) decorated CeO_2_ (Au–CeO_2_) with polymeric carbon nitride (PCN) via a modified thermal polymerization method. The combination of the interfacial interaction between PCN and CeO_2_ via N-O or C-O bonds, with the interior electronic transmission channel built by the decoration of Au NPs at the interface between CeO_2_ and PCN, endows CeAu–CN with excellent efficiency in the transfer and separation of photo-induced carriers, leading to the enhancement of photochemical activity. The amount-optimized CeAu–CN nanocomposites are capable of producing ca. 80 μmol· H_2_ per hour under visible light irradiation, which is higher than that of pristine CN, Ce–CN and physical mixed CeAu and PCN systems. In addition, the photocatalytic activity of CeAu–CN remains unchanged for four runs in 4 h. The present work not only provides a sample and feasible strategy to synthesize highly efficient organic polymer composites containing metal-assisted heterojunction photocatalysts, but also opens up a new avenue for the rational design and synthesis of potentially efficient PCN-based materials for efficient hydrogen evolution.

## 1. Introduction

With the future acceleration of environmental pollution and the energy crisis, photocatalytic water splitting has been the subject of intense research as it can replace fossil fuels by providing clean and renewable energy [1,2]. Following the pioneering work of Honda and Fujishima on a photoelectrochemical cell equipped with Pt-TiO_2_, considerable progress has been made in this field [3,4]. In particular, this is because photocatalysts play a vital role in water splitting systems; a large number of photocatalysts with appropriate band structure, visible light response and good photochemical stability have been designed and developed [5].

Polymeric carbon nitride (PCN), a new generation metal-free conjugated polymeric photocatalysts, has become a research hotspot for water splitting and CO_2_ reduction due to its unique advantages and significant physical and optical characteristics [6,7,8,9,10]. Given the potential application of PCN, improving the corresponding properties of materials is still an enormous challenge, since the methods of controlling the separation of photogenerated charge carriers are not well realized. To this end, developing a PCN-based photocatalytic system with higher solar-to-hydrogen energy (STH) conversion efficiency is one of the most promising ways to improve the separation and transfer of photogenerated charge carriers [11]. To date, many studies have proved that constructing reasonable and novel heterostructures can efficiently decrease exciton binding energy and enhance charge separation for PCN-based materials [12,13]. For example, recently, researchers have successfully constructed g-C_3_N_4_-based heterostructure composites, such as metal oxide–g-C_3_N_4_ [14,15], polymer–g-C_3_N_4_ [16] and sulfide–g-C_3_N_4_ [17,18] to enhance the photocatalytic activity through facilitating the separation of photogenerated charges and holes. In a typical example, Zheng et al. constructed a CdSe–CN S-scheme heterojunction via a linker-assisted hybridization approach, and showed that CdSe–CN exhibited superior photocatalytic reactivity to CdSe and CN in water splitting and CO_2_ conversion [19]. However, the efficiency of photogenerated carrier transportation has been limited owing to the interfacial effects between two different materials. Therefore, the rational design of PCN- based composites which would enable them to effectively adjust the interface charge transfer toward heterojunctions still is a huge challenge.

As a popular rare oxide, CeO_2_ nanomaterials possess good electrical conductivity, abundant redox chemistry, and plentiful surface oxygen vacancies due to the valence change between Ce^3+^ and Ce^4+^ oxidation states [20,21,22]. These excellent properties of CeO_2_ make it more likely to generate strong electron interaction with other materials, consequently, this will potentially improve the catalytic performance. For example, the Fe(OH)_3_–CeO_2_ composite, due to its tight interface effect, promoted the separation efficiency of photogenerated charges, thus showing excellent photocatalytic activity of water oxidation [23]. Additionally, Dong Lin’s group have undertaken a lot of related studies on CeO_2_–g-C_3_N_4_ complexes, which showed that the construction of CeO_2_–g-C_3_N_4_ heterojunctions provided an internal charge transport channel, leading to faster separation of electron–hole pairs and better photocatalytic activity than CeO_2_ and g-C_3_N_4_ [24,25,26]. Moreover, the properties of cerium can also be regulated by doping or acting as a carrier of various metals such as Ag, Pt, or Au, although the activity and selectivity of photocatalysts depend to a large extent on the type and concentration of dopants [27,28,29]. In a typical case, Primo et al. reported that the deposition of Au nanoparticles at low loading increases the photocatalytic activity of ceria more than the WO_3_ owing to its unique electronic structure [30]. Encouraged by the advantages of CeO_2_ and the excellent electrical conductivity of Au, the photocatalytic activity of CeO_2_-Au-PCN can be expected to be further improved. However, such three-component systems based on PCN have seldom been reported so far due to the great challenge in their synthesis.

Based on the above considerations, herein, novel Au NPs decorated CeO_2_ coupled with PCN ternary composites were designed and successfully synthesized via a modified thermal polymerization method, aiming to promote charge separation and improve the hydrogen evolution of PCN. FTIR and XPS results demonstrated a strong interfacial effect between PCN and CeAu through N-O or C-O bonds. The Au NPs can serve as a suitable electron acceptor and transfer channel, and CeO_2_ as a proper-level electron-accepting platform. Therefore, Au NPs provide a direct interior pathway to introduce photogenerated electrons from PCN into CeO_2_, which assists with the efficient transfer and separation of photogenerated electrons and holes. Not surprisingly, the ternary composites of CeO_2_–Au–PCN show strikingly ameliorated photocatalytic H_2_ evolution compared to the pristine and two-component CeO_2_–PCN system.

## 2. Results and Discussion

X-ray diffraction (XRD) was applied to phase analysis and crystal structure determination of samples. As presented in Figure 1, the diffraction peaks of pure CN located at 13.1° and 27.2° are attributed to the (100) and (002) planes, respectively. The diffraction peaks of CeAu at 2θ of 28.6°, 33.1°, 47.6°, 56.4°, 59.1°, 69.4°, 76.7°, 79.1° marked with “*” can be retrieved as well-crystallized face centered cubic structure of CeO_2_ JCPDS (no.34-0394) [31], whereas the other peaks at 38.2°, 44.4°, 64.8° marked with “•” can be assigned to the face-centered cubic structure of Au corresponding to (JCPDS no. 89-3697), indicating the formation of Au NPs on the surface of CeO_2_ [32]. For the CeAu–CN samples, diffraction peaks corresponding to CN, CeO_2_ and Au can be observed with the increasing CeAu content in composite, which confirms the successful formation of the three-component CeAu–CN system. The low intensity of the CeO_2_ and Au diffraction peaks can be observed owing to the low content of CeAu randomly distributed on the surface of CN. Nevertheless, compared with the pure CN, the (100) peak of x% CeAu–CN becomes much weaker, and almost disappears with the increasing CeAu content in composite. The phenomenon is similar to the previous report of disorders in the arrangement of in-plane structural motifs caused by doping oxygen atoms in CN [33]. Moreover, the gradual weakening and widening of the 27.2° peak is probably because of the structure fluctuation owing to the addition of CeAu.

The chemical structure information of the as-prepared materials can be well confirmed by FTIR spectra. In Figure 2, it can be seen that both pure CN and x% CeAu–CN composite materials exhibit similar vibrational modes of triazine heterocyclic ring molecular in polymeric carbon nitride. The characteristic peaks at 1200–1600 cm^−1^ belong to the stretching vibrations of CN heterocycles, while another significant characteristic peak at 810 cm^−1^ is caused by the breathing vibration of the triazine units. In addition, a weak broad peak at 2900–3300 cm^−1^ is attributed to the presence of the free amino group (e.g., NH_2_ or NH) and absorbed H_2_O molecules on the surface of the CN-based polymer. Surprisingly, from the partial magnification of FTIR spectra we can clearly see that a new weak band at ~985 cm^−1^ became more and more obvious with the increasing of CeAu content in x% CeAu–CN composites, probably due to the existence of the stretching vibration of the N-O group [34]. Because of this discovery, it is speculated that the O atom in CeO_2_ is hybridized with the N atom of CN, indicating the formation of the solid interfacial interaction between the CeAu and CN. The tight interfacial effect contributes to charge transfer, thereby improving the photocatalytic activity to some extent.

For the optical absorption properties of the samples to be investigated, the UV–Vis absorption measurement was carried out. As displayed in Figure 3a, CN shows an obvious absorption in the visible light region, while CeO_2_ exhibits optical absorption only in the ultraviolet light region (λ ≤ 400 nm), owing to the intrinsic nature of the samples. Notably, the visible light absorption of CeAu is significantly improved with marked absorption peaks at ca. 550 nm, which is known by common sense to be caused by the SPR effect of the loaded Au nanoparticles. Additionally, in Figure 3b, it can be seen that the optical edges gradually shift to the red wavelength region with the increase in the CeAu content in x% CeAu–CN compositions, but no significant light absorption caused by the SPR effect can be observed, which is mainly because the content of CeAu is too low compared with the urea precursor. Commonly, the band-gap energies of CN and CeO_2_ are calculated according to the following formula [35]:Ahv = (αhv − Eg)^n^(1)

Here, α is the absorbance, h is the Planck’s constant, v is the photon frequency, and Eg is the photonic energy band gap. Wherein, the value of n is related to the type of electronic transitions to which the samples belong. For CN belonging to the indirect bandgap, the value of n is 2, while for CeO_2_ belonging to the direct bandgap, the value of n is 0.5. Therefore, the band gap of CN and CeO_2_ are estimated from the Tauc plot (Figure 3a, inset) to be 2.65 eV and 3.20 eV, respectively.

The more detailed morphology and texture information of the samples were investigated by using scanning electron microscopy (SEM) and transmission electron microscopy (TEM). As shown in Appendix A, the SEM image 1.0% CeAu–CN sample exhibits a nanosheet with more porous holes and wrinkles relative to CN, which may benefit the increase in the BET surface areas of the sample. Meanwhile, Figure 4a distinctly shows some irregular CeAu nanoparticles randomly distributed on the surface of the CN photocatalyst. Together with Figure 4b, we can see the obvious lattice fringes with a d spacing of 0.31 nm are consistent with the value for the CeO_2_ (111) plane. With further enlargement of the selected area, the lattice-fringe spacing of 0.24 nm is observed, which is in agreement with the (111) plane of Au. Furthermore, the EDX element mapping images (Figure 4c) clearly display well-defined spatial distribution of C, N, Ce, O and Au for the 1.0% CeAu–CN photocatalyst. This results, together with XRD analysis, reveal the successful construction of the ternary CeO_2_-Au–CN hybrids. Notably, there is close contact between CeO_2_, Au, and CN, which is conducive to the charge transfer across their interface.

Having confirmed the morphology and structure information, we further examined the chemical compositions and element valence states of pristine CN, 1.0% CeAu–CN and CeAu materials by X-ray photoelectron spectroscopy (XPS). Not surprisingly, CN and 1.0% CeAu–CN show similar high-resolution C1s and N1s core-level XPS spectra. The C 1s spectra in Figure 5a can be resolved into three peaks located at ~284.6, 285.9, and 288.2 eV corresponding to the typical impurity carbon, carbon atoms in C-O, and sp^2^-hybridized carbon in N-containing aromatic ring (N-C=N), respectively. Moreover, the formation of C-O bonds may be related to oxygen-containing intermediates produced during the pyrolysis of urea or the lattice oxygen in the cerium oxide [36]. Figure 5b displays the XPS spectrum of N1s, which is mainly divided into four peaks, among which the peaks at 398.7 and 399.8 eV ascribed to sp^2^-hybridized nitrogen in the form of C=N-C and tertiary nitrogen N-(C)3 groups, respectively, and both of them are the main substructure units forming the tri-s-triazine heterocyclic ring. The other two peaks, located at 401.1 and 404.2 eV, are attributed either to the surface uncondensed C-N-H functional groups and charging effects or the positive charge localization in the heterocycles, respectively. Figure 5c shows the high-resolution XPS spectrum of Ce3d, by using a Gaussian fitting method, the Ce3d core level XPS fitted plot at ~885.4 and 903.6 eV can be assigned to 3d5/2 and 3d3/2 spin-orbit states, respectively, which ascertains the presence of both Ce^4+^ and Ce^3+^ in the CeAu, in agreement with previous studies [37,38]. Additionally, the weak peak of Ce3d can also be detected in the 1.0% CeAu–CN composites, though its content is shallow, which indicates the existence of Ce^4+^ and Ce^3+^ in the samples. Usually, the presence of Ce^3+^ is accompanied by the generation of oxygen vacancies (Ov), which play a vital role in enhancing visible light absorption as well as the interfacial interaction of ceria-based materials. Furthermore, the high resolution XPS spectra of O 1 s were displayed in Appendix A, for CeAu, the peak observed at 529.5 eV represents the lattice oxygen of CeO_2_. However, for 1.0% CeAu–CN composites, the O 1s spectra can be devolved into two peaks, one at 529.5 eV corresponds to the lattice oxygen of CeO_2_, another main peak at 531.7 eV is assigned to the C3-N+-O- species formed by the hybridization of CN and CeO_2_ or the O-H groups attach on the surface of the samples, which is consistent with the FTIR analysis results [39]. High-resolution XPS spectra of Au 4f orbital fitted to two peaks located at 84 and 87.7 eV corresponded Au4f7/2 and 4f5/2, respectively (Figure 5d), suggesting the presence of Au nanoparticles in CeAu. Interestingly, the binding energy of Au4f for 1.0% CeAu–CN is slightly lower than the CeAu sample, indicative of a strong interaction between Au and CN and CeO_2_. To some extent, this further confirms that there is close contact between the three, which probably facilitates the transfer of photo-induced carriers, enhancing the photocatalytic performance for photocatalysts.

Visible-light-induced photocatalytic H_2_ generation was then attempted by capitalizing on the prepared samples in the presence of 3 wt% Pt as a co-catalyst and 10 vol% triethanolamine (TEOA) as the sacrificial electron donor. As illustrated in Figure 6a, the CeAu has almost no photocatalytic activity of hydrogen production probably owing to the intrinsic property of CeO_2_. Of note, compared with CN, the photocatalytic activity of CeCN is slightly improved, while the CeAu–CN composites display compelling photocatalytic performance. The 1.0% CeAu–CN sample shows the fastest rate of H_2_ evolution driven by visible light (≈80 μmol h^−1^). As such, the materials of CeAu and CN just physically mixed (CeAu+CN) were selected as the reference and tested under the same reaction condition. Not surprisingly, there is no significant enhancement in catalytic activity. Therefore, the above experiments’ results jointly confirm that the outstanding photocatalytic performance may be ascribed to the intimate contact between the host and guest materials and the existence of gold nanoparticles, both of which are indispensable. In addition, as shown in Appendix A, it is observed that 1.0% CeAu–CN still exhibits the H_2_ evolution activity under extended wavelength irradiation, demonstrating that the CeAu strengthens the capture of visible light consistent with the analysis of DRS. In Figure 6b, the H_2_ evolution rates of the CeAu–CN ternary photocatalysts sensitively depend on the added amount of CeAu. The photocatalytic water reduction for the H_2_ evolution rate first increases and then decreases when the CeAu densities are varied. As the excessive accumulation of CeAu on the CN surface will build a light shielding effect, it may promote the recombination of photo-induced electrons and holes, thereby decreasing the photocatalytic efficiency. A comparison between the photocatalytic H_2_ evolution of our 1.0% CeAu–CN and that of other reported catalysts is listed in Table 1. It is worth noting that, compared with some photocatalysts, 1.0% CeAu–CN showed better photocatalytic performance in hydrogen production under visible light irradiation.

To evaluate the photostability of 1.0% CeAu–CN composites, the photocatalytic recycling performance of the sample was investigated under the same reaction conditions (Figure 6c). No obvious attenuation of hydrogen evolution rate (HER) is observed after the 16 h test, implying that 1.0% CeAu–CN ternary composition has excellent photostability. Moreover, the catalysts after the reaction were further subjected to characterization by XRD and FTIR (Appendix A), which reveals that there is no noticeable change in the crystal and chemical structure of the catalysts before and after the reaction. These observations explicitly elucidate the ternary CeAu–CN materials with the robust nature and high operation stability benefit for the photocatalysis of hydrogen evolution. Next, the best sample’s apparent quantum yield (AQY) was measured under various monochromatic light irradiation. As shown in Figure 6d, the AQY of H_2_ evolution matches well with the UV–Vis DRS spectra of the 1.0% CeAu–CN, confirming that the reaction is indeed driven by light irradiation, and the AQY value of 1.0% CeAu–CN is about 3.0% at 420 nm, which is higher than that of bulk CN previously reported.

Considering that the photocatalytic performance of photocatalyst is closely related to the charge–carrier separation and transfer process, we performed room-temperature photoluminescence (PL) and photoelectrochemical characterization. The room-temperature photoluminescence was conducted using 380 nm as the exciting wavelength. From Figure 7a, it is clear that all the photocatalysts show similar broad peaks centered at around 480 nm due to the band–band PL phenomenon, which is consistent with the results of DRS. At the same time, the PL emission intensities gradually decrease as the loading content of CeAu increases, which reveals the prohibited recombination of light-excited charge by the construction of heterojunctions. Time-resolved PL spectroscopy was utilized to probe the charge carrier dynamics of pristine CN and 1.0% CeAu–CN composite (Figure 7b). According to the three radiative lifetimes with different percentages, we can obtain the average lifetimes of pure CN and 1.0% CeAu–CN composites as 7.67 ns and 6.27 ns, respectively. Compared with CN, the lifetime of 1.0% CeAu–CN composite is relatively short, probably due to the internal transmission path constructed by Au NPs at the interface or the close interface contact between CN and CeO_2_, resulting in faster separation of photogenerated carriers. In addition, more charge carrier migration information was also reflected by the EPR analysis of CN and 1.0% CeAu–CN samples. In Appendix A, a single Lorentzian line centering at a g value of 2.003 is observed for CN and 1.0% CeAu–CN samples, indicating the presence of unpaired electrons on π-conjugated CN aromatic rings. Evidently, the EPR intensity of 1.0% CeAu–CN is greatly strengthened compared with CN. This strongly verifies that the modification of CeAu hybrids promotes the electron migration in the π-conjugated system of CN, possibly due to the intimated interface effects of CN and CeAu.

The electron transfer behavior can also be further demonstrated in the following electrochemical experiments. Firstly, the interface charge transfer resistance of the electrons was implemented via an electrochemical impedance spectroscopy (EIS) test. The experimental results show that the 1.0% CeAu–CN photocatalyst possesses a smaller high-frequency semicircle than the pure CN (Figure 7c), meaningfully indicating the lower charge-transfer resistance in a hybrid that ensures faster electrons trainer. Next, the interface charge separation kinetics of the samples can also be reflected by the photocurrent density-time response plot. As displayed in Figure 7d, the photocurrent drastically increases and decreases, respectively, when the power supply is switched on and off. Only about 0.5 μA cm^−2^ of photocurrent density emerged for pure CN, whereas 1.2 μA cm^−2^ of photocurrent density is generated for 1.0% CeAu–CN electrode, implying a lower recombination rate of electron–hole pairs and good electrical conductivity for the 1.0% CeAu–CN electrode compared to the pristine one. Therefore, combined with the PL analysis and the results of photo/electrochemical studies, it is well established that the modification of CeAu successfully improved the separation and migration of light-generated charges compared with the pristine CN. The enhanced photogenerated charge separation of CeAu–CN is responsible for improving its photocatalytic activity.

In order to further investigate the transfer process of photogenerated charge, it is urgent to obtain the relative band position of CN and CeO_2_. Therefore, in the next experiment, we carried out an electrochemical analysis of CN and CeO_2_ to evaluate their electronic band structure. Figure 8a,b show the positive slope of the plot indicating both CN and CeO_2_ are n-type semiconductors, and their corresponding flat potential is calculated to be −1.2 and −0.69 V reference the saturated calomel electrode (SCE), which are equivalent to −0.96 and −0.45 V versus the normal hydrogen electrode (NHE), respectively. As previously established, for the n-type semiconductor, the actual conduction band is 0~0.1 eV higher than the flat potential due to the electron effective mass and the carrier concentration [44,45,46]. Here, the voltage difference between the conduction band and the flat potential is set to 0.1 eV. According to the electrochemical results and the related calculations, the bottom conduction bands for CN and CeO_2_ are −1.06 and −0.55 eV, while the corresponding valence bands are 1.59 and 2.35 eV, respectively.

Based on the results and analysis of the previous characterizations and the above relative energy band levels of CN and CeO_2_, a plausible mechanism schematic of photogenerated charge transfer and separation and photochemical reactions of CeAu–CN ternary composites driven by the light is proposed. As displayed in Figure 9, under visible light irradiation, it is acceptable that the photogenerated electrons of CN transfer to the CB of CeO_2_ with the assistance of Au NPs. Because of its strong “electron sink” effect, Au NPs can be served as carrier conductors, providing an interior direct channel to facilitate the separation and transport of photo-induced carriers at the interface of type-ΙΙ heterostructure [47,48,49]. The icing on the cake is that the tight interface between CN and CeO_2_ also provides a good platform for charge transfer. As a result, the photogenerated charges of PCN are well separated and transferred for a more efficient reaction with target reactants, so its photocatalytic hydrogen production performance is greatly improved.

## 3. Materials and Methods

### 3.1. Materials

All chemicals used in the synthesis were of analytical grade without further purification. Cerium nitride hexahydrate (Ce(NO_3_)_3_·6H_2_O), sodium hydroxide (NaOH), sodium sulfate (Na_2_SO_4_) and urea purchased from Sinopharm Chemical Reagent Co. Ltd., Shanghai, China. Chloroauric acid (HAuCl_4_·4H_2_O) and chloroplatinic acid(H_2_PtCl_6_·6H_2_O) were supplied by Alfa Aesar China Co., Ltd. (Tianjin, China).

### 3.2. Synthesis

#### 3.2.1. Synthesis of CeO_2_

As previously reported, CeO_2_ nanomaterials were synthesized by the hydrothermal method using cerium nitride hexahydrate (Ce(NO_3_)_3_·6H_2_O) as the precursor [50]. To describe the experiment briefly, first, 4 mmol of Ce(NO_3_)_3_·6H_2_O was dissolved in 10 mL water, subsequently, 70 mL of 6 M NaOH solution was added to the above solution dropwise and continuously stirred at room temperature for 2 h. Then, the gray mud was transferred into a 100 mL Teflon-lined stainless-steel autoclave and heated at 120 °C for 24 h under autogenous pressure. After cooling to room temperature, the gray precipitates were collected by centrifugation with deionized water and ethanol many times, followed by drying at 80 °C overnight in an oven, resulting in CeO_2_.

#### 3.2.2. Synthesis of Au–CeO_2_

Au-loaded CeO_2_ was prepared via a simple impregnation method using aqueous HAuCl_4_·4H_2_O solution [51]. The detailed preparation procedure employed is described below. Firstly, 0.3 g CeO_2_ was dispersed in 15 mL of deionized water to obtain uniform suspension by stirring for 0.5 h, then added a certain amount of HAuCl_4_·4H_2_O solution, and the mixture was stirred for another 1 h. The lavender powder was formed after dried at 80 °C for 16 h under vigorous magnetic stirring. In this paper, the Au–CeO_2_ sample with theoretical 1.5 wt% Au was obtained and denoted as CeAu.

#### 3.2.3. Synthesis of CeAu–CN, Ce–CN and CN

The CeAu–CN photocatalysts were obtained through a modified thermal polymerization of urea molecules with certain amount of CeAu similar to the experimental procedure described by our group [52]. Typically, 10 g urea mixed with different amount of CeAu in 20 mL deionized water with vigorous magnetic stirring at room temperature for 2 h and then stirring at 85 °C to remove water. Afterwards the resultant solids were ground and calcined at 550 °C for 2 h in N_2_ with the speed of 4.6 °C min^−1^ to obtain the final samples. There were noted as x% CeAu–CN, where x (x = 0.5, 1, 1.5, 2.0) is the percentage weight content of CeAu to urea. The Ce–CN composite was prepared by the same procedure of CeAu–CN, only with the CeAu replaced by CeO_2_. The pure PCN (denoted as CN) was prepared by the same method without adding the CeAu.

### 3.3. Characterizations

The crystal structure of the samples was determined with the support of Powder X-ray diffraction (XRD) which was executed on Bruker D8 Advance focus using Cu Kα1 radiation and recorded in the 2θ rang 10–80°. Fourier transform infrared (FTIR) spectra were measured on a Nicolet-6700 in the frequency range of 4000–400 cm^−1^. The morphology of the as-made samples was investigated by field emission scanning electron microscopy (SEM) (JSM-6700F) and FEI Tecnai20 transmission electron microscopy (TEM). Brunauer–Emmett–Teller (BET) specific surface area and porosity of photocatalysts were carried out by sorption using Micromeritics ASAP 2010 instrument at 77 K. The compositions and chemical valence states of as-synthesis samples were collected from XPS spectra analyzer using a Thermo ESCALAB250 instrument with a monochromatized Al Kα line source (200 W). The optical properties of the materials can be demonstrated by the ultraviolet-visible diffuse reflectance spectra (UV–Vis DRS) measured on a Varian Cary 500 Scan UV–Vis system using BaSO_4_ as a reference. In addition, photoluminescence (PL) spectra obtained on an Edinburgh FI/FSTCSPC 920 spectrophotometer using 380 nm as the exciting wavelength. Electron paramagnetic resonance (EPR) measurements were performed on a Bruker model A300 spectrometer. The BioLogic VSP-300 electrochemical system was used to measure the electrochemical performance of samples in a traditional three-electrode cell with a Pt plate and saturated calomel electrode (SCE) as counter electrode and reference electrode, respectively.

### 3.4. Photocatalytic Activity Test

The photocatalytic hydrogen evolution experiment was carried out in an online reaction system. The schematic of photocatalytic water splitting reaction is shown in Appendix A, which mainly includes three parts: photocatalytic water splitting reaction system, vacuum circulation system and analysis and testing system. The detailed experimental process is as follows: the powder photocatalyst (50 mg) was suspended in an aqueous solution (100 mL) containing 10 mL triethanolamine as the holes’ sacrificial reagent. The 3 wt% Pt co-catalyst was loaded onto the surface of photocatalysts by in situ photo-deposition method using H_2_PtCl_6_ as precursor during the reaction. The reaction temperature was always maintained at 12 °C by the circulating condensing equipment. Subsequently, the reaction system was sealed and evacuated many times to completely remove the air, which was then irradiated under a 300 W Xe lamp using a 420 nm cutoff filter. The different wavelength experiments were similar to previous ones, except that the cutoff filters were changed to a different cutoff wavelength. The generated hydrogen was determined by a gas chromatograph equipped with a thermal conductive detector (TCD) and a 5 Å molecular sieve column with high-purity argon as the carrier gas.

The apparent quantum yield (AQY) for H_2_ evolution was measured using monochromatic LED lamps with band pass filter of 380, 405, 420, 450, 470, 495 nm. The irradiation area was controlled as 3 × 3 cm^2^. The AQY was calculated based on the following formula: AQY = Ne/Np × 100 = 2MNAhc/SPtλ × 100%, where Ne is the amount of electrons involved in the reaction, Np is the amount of incident photons, M is the quantity of hydrogen molecules produced by reaction, NA is Avogadro constant, h is the Planck constant, c is the speed of light, S is the irradiation area, P is the intensity of the irradiation, t is the photo-irradiation time, and λ is the wavelength of the monochromatic light.

## 4. Conclusions

In summary, a three-component CeAu–CN heterostructure has been successfully established via a modified thermal polymerization method. Owing to the interfacial interaction between PCN and CeO_2_ via N-O or C-O bands and the interior electronic transmission channel constructed by the decorated of Au NPs at the interface, CeAu–CN has been confirmed to be highly efficient in the separation and transfer of photogenerated carriers, and greatly enhanced photocatalytic activity. The amount-optimized 1.0% CeAu–CN nanocomposite exhibits the highest H_2_ evolution rate of 80.1 μmol h^−1^ under visible light irradiation (λ > 420 nm), and the photocatalytic activity of CeAu–CN still remains unchanged for four runs in 4 h. More importantly, the present study not only discloses the importance of interfacial effects on photocatalytic activity, but also opens a promising avenue for the rational design and fabrication of heterogeneous interface-containing metals for highly effective solar water splitting.

## Figures and Tables

**Figure 1 molecules-27-07489-f001:**
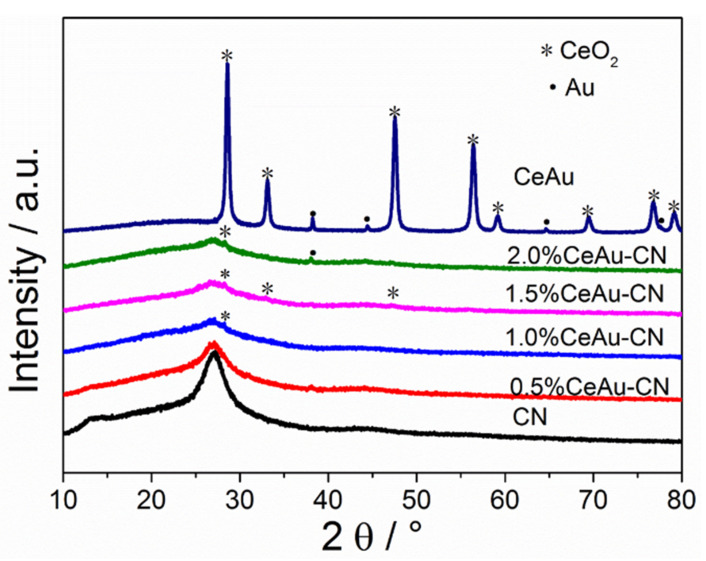
The XRD patterns of CN, x% CeAu–CN (x = 0.5, 1.0, 1.5, 2.0), and CeAu.

**Figure 2 molecules-27-07489-f002:**
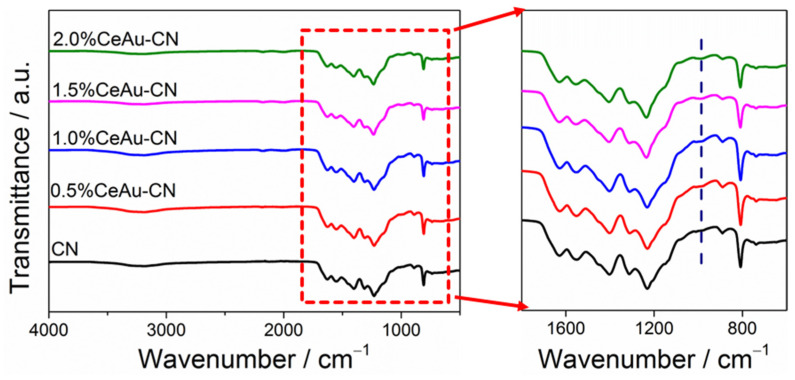
FTIR spectra and the partial magnified graph of CN and x% CeAu–CN (x = 0.5, 1.0, 1.5, 2.0).

**Figure 3 molecules-27-07489-f003:**
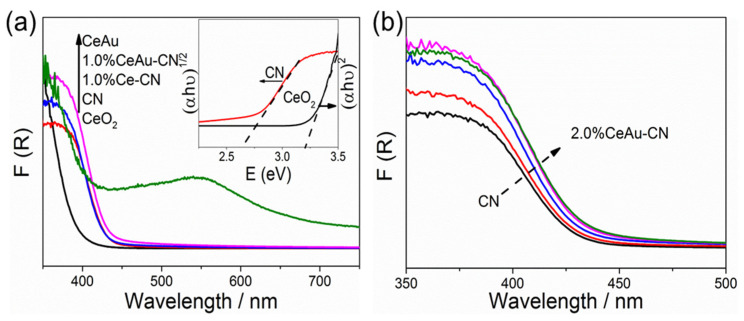
(**a**,**b**) UV–Vis DRS spectra of CN, CeO_2_, CeAu, Ce–CN and x% CeAu–CN samples (x = 0.5, 1.0, 1.5, 2.0).

**Figure 4 molecules-27-07489-f004:**
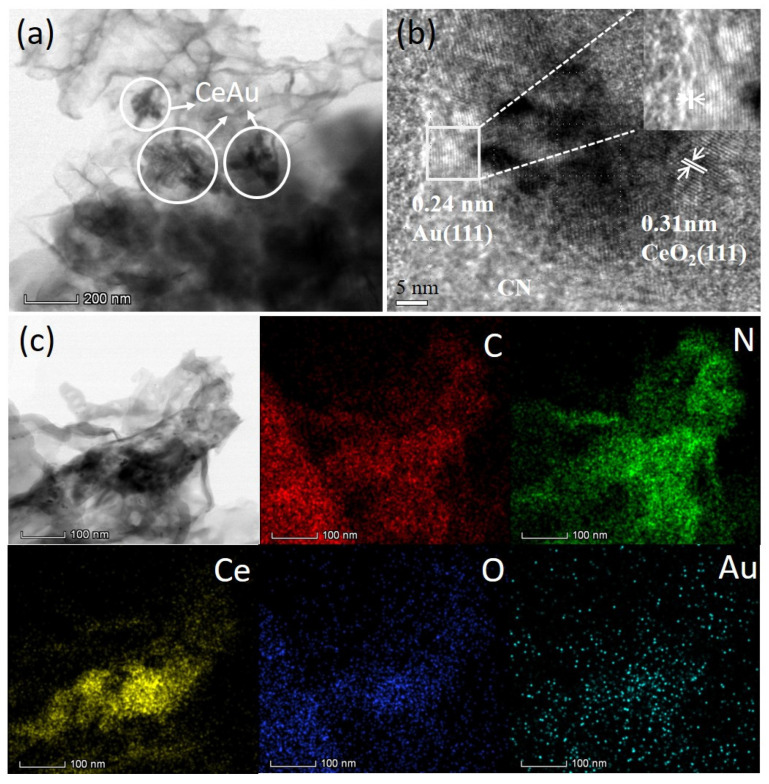
(**a**) TEM and (**b**) HRTEM of 1.0% CeAu–CN; (**c**) STEM image and element mapping images of C, N, Ce, O and Au for the 1.0% CeAu–CN photocatalyst.

**Figure 5 molecules-27-07489-f005:**
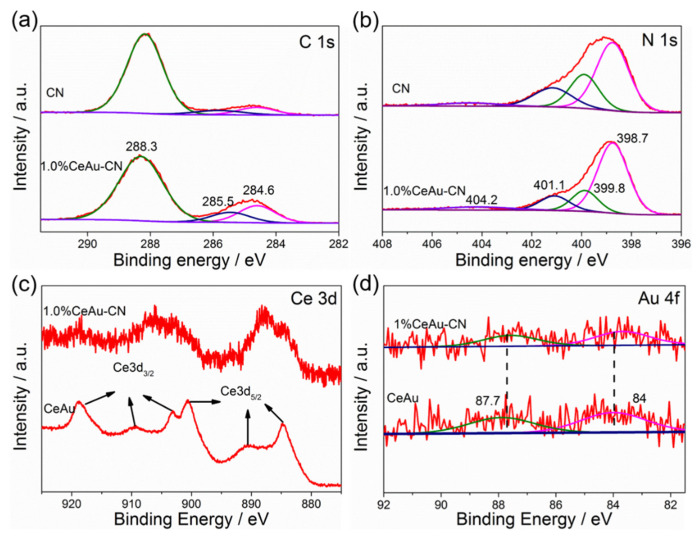
High-resolution XPS spectra of (**a**) C1s region and (**b**) N1s region of the CN and 1.0% CeAu–CN composite; (**c**) Ce3d region and (**d**) Au4f region of the 1.0% CeAu–CN and CeAu.

**Figure 6 molecules-27-07489-f006:**
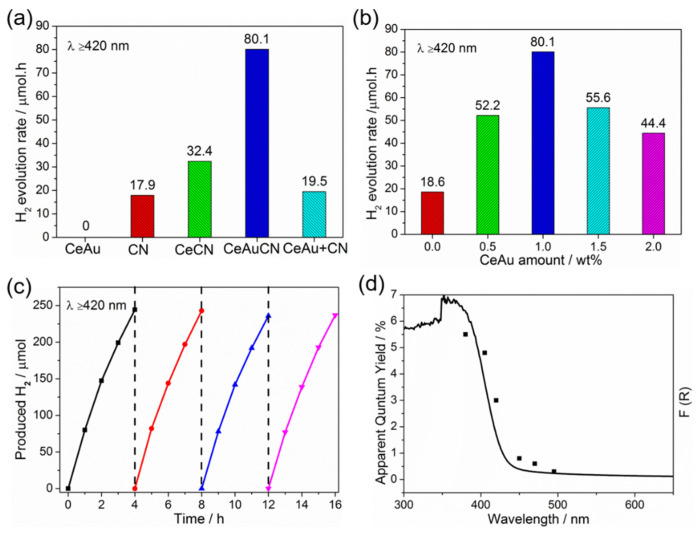
(**a**) Photocatalytic performance for H_2_ production of CeAu, pure CN, CeCN, CeAuCN and CeAu+CN; (**b**) Photocatalytic activity for H_2_ evolution rate of modified CN with different amounts of CeAu under visible light irradiation (λ ≥ 420 nm); (**c**) The photocatalytic stability of the 1.0% CeAu–CN; (**d**) Wavelength dependence of AQY on the H_2_ evolution in the 1.0% CeAu–CN sample.

**Figure 7 molecules-27-07489-f007:**
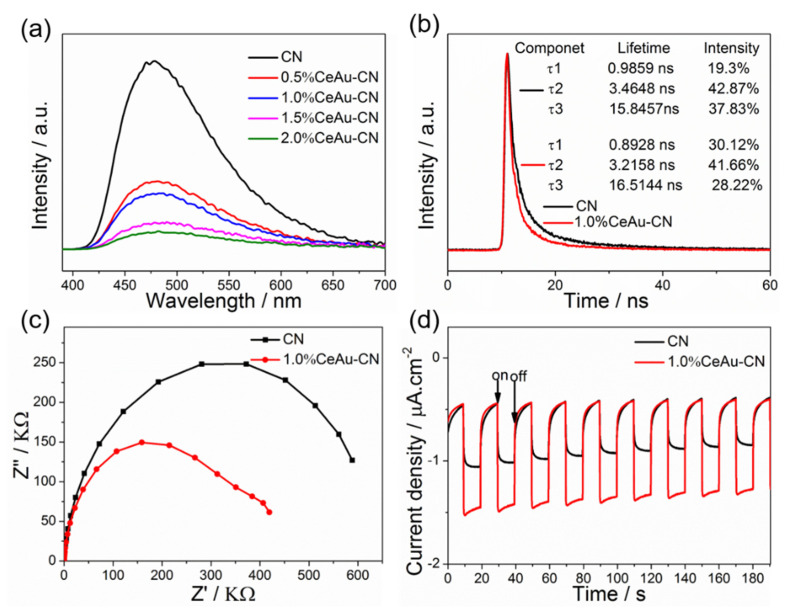
(**a**) PL spectra and (**b**) Time-resolved PL spectra of CN and x% CeAu–CN (x = 0.5, 1.0, 1.5, 2.0) samples; (**c**) EIS tests and (**d**) Photocurrent response of CN and 1.0% CeAu–CN.

**Figure 8 molecules-27-07489-f008:**
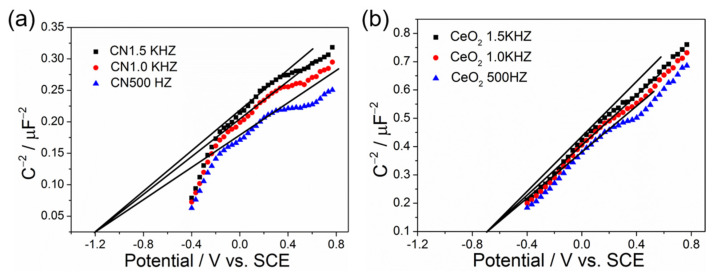
Mott–Schottky plots of (**a**) CN and (**b**) CeO_2_ film electrodes at different frequency in 0.2 M Na_2_SO_4_ aqueous solution (pH = 6.8).

**Figure 9 molecules-27-07489-f009:**
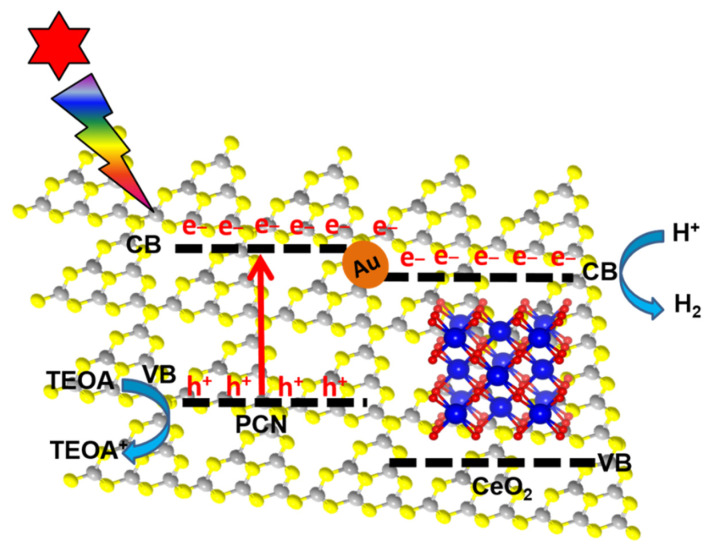
Schematic illustrations of the possible transfer channels of electron–hole pairs separation.

**Table 1 molecules-27-07489-t001:** Comparison between the photocatalytic activity of 1.0% CeAu–CN and that of other reported catalysts for photocatalytic hydrogen evolution.

Photocatalysts	Reaction Conditions	Light Source	H_2_ Production (μmol h^−1^ g^−1^)	Ref.
1.0% CeAu–CN	0.05 g catalyst, 3 wt%Pt, TEOA solution (10%)	300 W Xe lamp λ > 420 nm	1602	This work
CeO_2_–g-C_3_N_4_	0.05 g catalyst, 3 wt%Pt, TEOA solution (10%)	300 W Xe lamp λ > 420 nm	1100	[25]
CeO_2_–g-C_3_N_4_	0.05 g catalyst, 0.5 wt%Pt, lactic acid solution (20%)	300 W Xe lamp λ > 420 nm	73.12	[40]
N-CeOx–g-C_3_N_4_	0.05 g catalyst, 1 wt%Pt, TEOA solution (10%)	300 W Xe lamp λ > 420 nm	292.5	[41]
Au–SnO_2_–g-C_3_N_4_	0.1 g catalyst, methanol solution (20%)	300 W Xe lamp λ > 400 nm	770	[42]
g-C_3_N_4_–Au–C-TiO_2_	0.01 g catalyst, TEOA solution (10%)	300 W Xe lamp λ > 420 nm	129	[43]

## Data Availability

Not applicable.

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
