# Peer review of "Modification of Polymeric Carbon Nitride with Au–CeO2 Hybrids to Improve Photocatalytic Activity for Hydrogen Evolution"

_molecules, 2022, doi:10.3390/molecules27217489_

Round 1

Reviewer 1 Report

The paper titled “Modification of Polymeric Carbon Nitride with Au/CeO2 Hybrids to Improve Photocatalytic Activity for Hydrogen Evolution” by Zhang et al. The manuscript was on the use of Au  nanoparticles (NPs) decorated CeO2 (Au/CeO2) with polymeric carbon nitride (PCN) for efficient hydrogen evolution under visible light irradiation. The photocatalytic activity of CeAu-CN maintains unchanged for 4 runs in 4 h.

I believe after minor revisions. The manuscript can be considered for publishing in Molecules

Line 19 Page 1: built by the decorated of Au NPs, change decorated to decoration.

Introduction lines 33 – 35: it is not clear it needs to be restructured

Line 35: pioneers should be deleted.

Line 48 (STH) should be after immediately after solar-to-hydrogen

Line 75 is not clear

The objective of the research should be clearly stated in the last paragraph of the introduction section.

Authors should include a table comparing the results obtained with literatures

More information should be provided in the introduction indicating the relevance of the study.

Conclusion should be supported by results obtained.

The manuscripts need to be given to a native English speaker for proper editing.

Author Response

Thanks for the reviewer’s comments very much! We have taken into account the detailed suggestions presented below and carefully revised this manuscript. Please see the attachment.

Reviewer 2 Report

This manuscript describes a novel polymeric CN doped with Au/CeO2 to achieve the high photocatalytic activity. The concept of Au/CeO2/g-C3N4 has been a hot topic in recent years, and such exploration is certainly meaningful and interesting. However, this manuscript seems to focus a lot on the characterization of the material itself, rather than what the authors have claimed in the title, the enhanced photocatalytic activity. I suggest that the authors elaborate more on the photocatalytic activity part, or consider changing the title.

Although I agree that the authors have done an outstanding job in characterizing the CeAu-CN with different doping percentages, I am genuinely confused about the photocatalytic discussion. As the authors state, they used an online reaction system, does this mean that it's a computer simulation instead of an actual experiment? If so, I believe the actual experiment needs to be conducted to verify the findings, otherwise it's only half done. If the authors indeed conducted the experiment, a lot more detailed discussion should be included to talk about its photocatalytic activity.

Author Response

We will be happy to edit the text further, based on helpful comments from the reviewer. We have adjusted the order of some contents and addede the diacussion on photocatalytic activity in the revised manuscript. We sincerely hope that our logic is now easier to focus on the improvement of photocatalytic activity. Please see the attachment.

Round 2

Reviewer 2 Report

I would like to first thank the authors for addressing my concerns and answering my questions regarding the manuscript. I was confused about the photocatalytic experiment conducted but after the explanation, it is now easier to understand.

After reviewing this revised manuscript, I believe that I am now much more confident about the authors' claim of the title regarding the photocatalytic activity. Specifically, I like the portion where the authors cross-compare state-of-the-art research and show excellent H2 production. The added portion in the discussion section makes the manuscript thorough and suitable for publication.

My only recommendation is that the authors include Fig R1 and its discussion in the supplementary material, or in materials and methods. I did not see it in the text nor in the supplementary.

Author Response

Thanks for the reviewer’s kind suggestion. I have made corresponding modifications in the revised manuscript and the supplementary material. Please see the attachment
